# Helicase-Like Transcription Factor HLTF and E3 Ubiquitin Ligase SHPRH Confer DNA Damage Tolerance through Direct Interactions with Proliferating Cell Nuclear Antigen (PCNA)

**DOI:** 10.3390/ijms21030693

**Published:** 2020-01-21

**Authors:** Mareike Seelinger, Marit Otterlei

**Affiliations:** Department of Clinical and Molecular Medicine, Faculty of Medicine and Health Sciences, Norwegian University of Science and Technology NTNU, N-7491 Trondheim, Norway; mareike.seelinger@ntnu.no

**Keywords:** Translesion synthesis (TLS), mutagenicity, DNA damage tolerance (DDT), APIM, RAD5

## Abstract

To prevent replication fork collapse and genome instability under replicative stress, DNA damage tolerance (DDT) mechanisms have evolved. The RAD5 homologs, HLTF (helicase-like transcription factor) and SHPRH (SNF2, histone-linker, PHD and RING finger domain-containing helicase), both ubiquitin ligases, are involved in several DDT mechanisms; DNA translesion synthesis (TLS), fork reversal/remodeling and template switch (TS). Here we show that these two human RAD5 homologs contain functional APIM PCNA interacting motifs. Our results show that both the role of HLTF in TLS in HLTF overexpressing cells, and nuclear localization of SHPRH, are dependent on interaction of HLTF and SHPRH with PCNA. Additionally, we detected multiple changes in the mutation spectra when APIM in overexpressed HLTF or SHPRH were mutated compared to overexpressed wild type proteins. In plasmids from cells overexpressing the APIM mutant version of HLTF, we observed a decrease in C to T transitions, the most common mutation caused by UV irradiation, and an increase in mutations on the transcribed strand. These results strongly suggest that direct binding of HLTF and SHPRH to PCNA is vital for their function in DDT.

## 1. Introduction

Cells are constantly exposed to endogenous and exogenous agents that cause DNA lesions. DNA translesion synthesis (TLS), fork reversal, and template switch (TS) are DNA damage tolerance (DDT) mechanisms handling DNA lesions or other obstacles during replication. This is achieved by one or more specialized low-fidelity polymerases in TLS (TLS polymerases), by converting the replication fork into a stabilized chicken-foot structure intermediate, or by using the nascent strand in the sister chromatid as a template in TS. These mechanisms allow cells to continue replication and to prevent replication fork collapse [1].

One day of sun exposure is estimated to induce up to 10^5^ UV-induced photolesions per cell [2]. UVB irradiation is absorbed by the DNA and induces mainly two types of UV-photodimers: cyclobutane pyrimidine dimers (CPDs) and pyrimidine pyrimidone (6-4)photoproducts (6-4PPs). CPDs account for the largest fraction of all photoproducts [3,4]. They are the major mutagenic UV lesion in mammalian cells, because they exhibit a slower repair-rate and are therefore often bypassed by TLS polymerases [5,6,7]. However, 6-4PPs, on the other hand, are bulky lesions which are rapidly detected and repaired by nucleotide excision repair (NER) [8]. The Xeroderma Pigmentosum variant (XPV) syndrome is associated with POLη deficiency, but has intact nucleotide excision repair (NER) [9]. XPV patients are hypersensitive toward sunlight and have an ~1000 times higher skin cancer incidence rate [10,11,12], illustrating the importance of properly regulated TLS, even in presence of functional NER (reviewed in [13]). 

The DDT mechanisms are at least partly coordinated by mono- and polyubiquitination of K164 of PCNA (reviewed in [14]). The RAD5 homologues, HLTF and SHPRH, are ubiquitin ligases containing a RING domain which is involved in polyubiquitination of PCNA [15]. They are therefore believed to be important for inducing TS. In addition, HLTF mediates fork reversal via its HIRAN domain interacting with the 3’ends of a frayed fork [16]. For SHPRH, which lacks a HIRAN domain, fork reversal has not been reported yet. Human cells have at least two additional important fork reversal proteins, SMARCAL1 and ZRANB3. The latter contains a functional PCNA interacting sequence, AlkB homolog 2 PCNA interacting motif (APIM) [17]. In addition, SMARCAL1, ZRANB3, and HLTF can restore a three-way-replication fork from a four-way-reversed replication fork [16,18]. Furthermore, both RAD5 homologs are reported to be involved in TLS: HLTF by recruiting POLη after UV-induced DNA damage, and SHPRH by recruiting POLκ after methyl methanesulfonate (MMS)-induced DNA damage [19]. HLTF and SHPRH are regarded as candidate tumor suppressor genes, because loss of function or dysregulation has been linked to cancer development [20,21,22], and HLTF is often epigenetically silenced by promotor hypermethylation in colon cancers (~40%) [23]. Interestingly, both RAD5 homologs contain potential PCNA interacting motifs, APIMs, within the helicase domain at their C-terminus; KFIVK (amino acid (aa) 959–963) in HLTF and RFLIK (aa 1631–1635) in SHPRH.

Serving as a scaffold protein, PCNA switches between protein interaction partners, using either of the two motifs: the PIP-box or APIM. While the majority of APIM-containing proteins are involved in various cellular stress responses, including DNA repair, TS and TLS, the PIP-box is found in multiple proteins essential for replication [24,25,26]. The two motifs share the same interaction site on PCNA, which is a potential binding site for over 600 proteins [27,28,29,30]. Interactions with PCNA are therefore coordinated at multiple levels, including affinity-driven competitive inhibition, context/cellular state and posttranslational modifications on either PCNA or the interacting proteins [17,31]. Data suggest that cellular stress, e.g., such as replication stress, is important for increased APIM–PCNA affinity [17,24,26,27,32,33,34]. Indeed, functional APIM is verified in three proteins directly involved in DDT, ZRANB3 [17], TFII-I [24,35], and the catalytic subunit of POLζ, REV3L [36], and in two proteins involved in the regulation of TS, FBH1 [28], and RAD51B [24]. Here we show that APIM in both HLTF and SHPRH interact with PCNA, and that the binding of HLTF and SHPRH to PCNA is important for their function in DDT after UV-induced DNA damage. 

## 2. Results and Discussion

### 2.1. APIM in HLTF Is a Functional PCNA Interacting Motif

APIM in HLTF (KFIVK) fused to CFP and co-expressed with HcRed-tagged PCNA colocalized with PCNA in foci resembling replication foci (Figure 1A, mid two images). We have previously shown that a mutation of the second position in APIM from an aromatic amino acid to alanine reduced the affinity to PCNA by ~50% [37]. Accordingly, the HLTF APIM F2A mutant (K**A**IVK) fused to YFP did not colocalize with PCNA in foci when co-expressed with K**F**IVK-CFP and HcRed-PCNA (Figure 1A, left image), a first indication for the importance of APIM in HLTF. However, both wild type and APIM mutant versions (F960A) of full-length YFP-HLTF colocalized with overexpressed HcRed-PCNA (Figure 1D,H). The persistent colocalization of YFP-HLTF F960A with PCNA in replication foci indicates that the mutation in APIM only reduces, but does not abolish the interaction with PCNA, and that other proteins partners participating in the same complex may interact with PCNA. The cellular localization of YFP-HLTF F960A was, however, slightly different from the wild type protein, i.e., we detected a higher level of fluorescence in the cytosol. Thus, the mutation in APIM of HLTF might have reduced the stability, reduced nuclear retention, or increased nuclear export of the protein (Figure 1B, quantified in C). Therefore, we next tested if the nuclear level of HLTF is actively regulated. The nuclear HLTF level was measured after overexpression and treatment of the cells with Leptomycin B (an active nuclear export blocking drug). We found increased nuclear levels of both YFP-HLTF and YFP-HLTF F960A (Figure 1C). This indicates that the level of HLTF in the nucleus is actively regulated, but that the export of HLTF is independent of APIM and thus independent of a direct HLTF-PCNA interaction (Figure 1C). 

To further investigate the importance of APIM in HLTF for colocalization with PCNA in replication foci, HLTF and HLTF F960A were overexpressed together with APIM-peptides and PCNA. The intensity of YFP-HLTF in PCNA foci was significantly reduced by co-expression of KFIVK-CFP (APIM in HLTF), and an even stronger reduction could be achieved by overexpression of RWLVK-CFP, an APIM-version with increased PCNA-affinity [36] (Figure 1E,F, quantified in G). The intensity of YFP-HLTF F960A in PCNA foci was initially stronger than YFP-HLTF; however, after overexpression of APIM-peptides (KFIVK-CFP or RWLVK-CFP), foci intensity was reduced to the same or lower level as measured for YFP-HLTF (Figure 1I,J, quantified in K). These results show that localization of both wild type HLTF and HLTF F960A in PCNA foci are reduced by overexpression of peptides containing the APIM sequence of HLTF, supporting that APIM in HLTF is a functional PCNA interacting motif. 

### 2.2. Nuclear Localization of SHPRH Depends on Its Interaction with PCNA 

Like HLTF, APIM in SHPRH (RFLIK) was fused to CFP and co-expressed with HcRed-tagged PCNA. R**F**LIK-CFP colocalized with PCNA, while the F2A APIM mutant version (R**A**LIK-CFP) did not (Figure 2A). The same APIM mutation in full-length SHPRH (F1632A), led to a strong reduction in nuclear localization compared to wild type SHPRH (Figure 2B, quantified in C). These results could suggest that the interaction with PCNA is necessary for nuclear localization of SHPRH or that the mutant SHPRH protein is less stable. To explore if the nuclear localization of SHPRH is dependent on a direct interaction with PCNA, we examined if the fraction of nuclear SHPRH can be reduced by treatment with an APIM containing cell penetrating peptide (APIM-peptide), which has earlier been shown to block the binding of APIM-containing proteins to PCNA [27]. Indeed, the fluorescence intensity of GFP-SHPRH in the nucleus was reduced upon APIM-peptide treatment (Figure 2C), and this effect was not achieved by treatment with a mutant version of the APIM-peptide with reduced affinity for PCNA (MutAPIM-peptide, W2A) [37]. Together, these results indicate that the nuclear localization of SHPRH is dependent on its direct binding to PCNA via APIM. 

Like HLTF, both GFP-SHPRH and GFP-SHPRH F1632A colocalized with overexpressed HcRed-PCNA in replication foci (Figure 2D,E). Thus, the residual affinity of SHPRH F1632A to PCNA and/or its interaction with other PCNA interacting proteins is sufficient to cause the observed localization of SHPRH F1632A in replication foci. 

### 2.3. APIM in SHPRH and HLTF Is Required for Maximal Pull Down of PCNA After DNA Damage

Our results suggest that APIM in both HLTF and SHPTH are functional PCNA-interacting motifs. To further verify this, we examined if endogenous PCNA can be pulled down by the GFP-SHPRH and YFP-HLTF fusion proteins. As APIM–PCNA interactions are relatively weak, and PCNA-interactions with HLTF and SHPRH are expected to be dynamic and to increase upon DNA damage, we treated the cells with the DNA alkylating agent MMS and gently cross-linked the cells with formaldehyde before making extracts (see Material and Methods and [24]). We could not detect any specific PCNA pull down in absence of MMS treatment (data not shown), but both YFP-HLTF and GFP-SHPRH pulled down more endogenous PCNA than the YFP control (Figure 2F, normalized against total GFP pulled down). The mutated APIM versions of the proteins, especially SHPRH F1632A, pulled down less PCNA compared to the wild type proteins. We detected less full-length protein and more degraded protein of the APIM mutant versions compared to the wild type proteins (Appendix A), particularly of HLTF F960A, and this may possibly also influence the level of PCNA pull down. However, increased degradation of the APIM mutant versions of HLTF and SHPRH may also support that their interaction with PCNA is important for the stability of these proteins. 

### 2.4. Direct Interaction with PCNA Is Important for the Regulation of DDT by HLTF and SHPRH

To further examine the functionality and the impact of APIM in HLTF and SHPRH in DDT, we performed SupF mutagenesis assays. We measured the mutation frequency and analyzed the mutation spectra in UV-damaged reporter plasmids in cells overexpressing wild type or APIM mutant versions of HLTF or SHPRH. Overexpression of HLTF did not significantly change the mutation frequency compared to control, although a tendency toward a reduction was observed. However, overexpression of HLTF F960A significantly increased the mutation frequency by 45%, compared to both control and HLTF wild type overexpressing cells (Figure 3A). This suggests that a direct interaction with PCNA is important for HLTF’s ability to support error-free repair. One of the most studied TLS polymerases, POLη, is important in bypassing UV lesions and essential for error-free bypass of TT-CPDs [38]. HLTF has previously been reported to stimulate bypass by POLη [19]. Thus, our results could indicate a lack of POLη stimulation in HLTF F960A overexpressing cells and that POLη stimulation by HLTF is dependent on a direct interaction with PCNA. Recently, a loss-of-function mutation in HLTF has been reported to increase the amount of DNA damage due to a decrease in PCNA polyubiquitination [39]. Thus, the increase in mutation frequency in presence of HLTF F960A overexpression could also be caused by a reduction of PCNA polyubiquitination and TS, indicating that APIM in HLTF is important for PCNA polyubiquitination mediated by HLTF. Recently, Masuda and colleagues reported that a HLTF–PCNA interaction at stalled primer ends reduced the polyubiquitination activity of HLTF in vitro, and that the polyubiquitination activity of HLTF could be partly restored by a mutation in the putative APIM in HLTF (F960A) [40]. However, they did not know if APIM was a functional PCNA interacting motif. Here we show that APIM in HLTF is a functional PCNA interacting motif; thus, our results support that the restoration of HLTF’s polyubiquitination activity, seen by Masuda and colleagues, was due to a reduced PCNA interaction. Therefore, the increased mutation frequency in HLTF F960A overexpressing cells detected in our study is probably not caused by reduced, but rather by increased polyubiquitination and excessive fork reversal. The latter is associated with genomic instability [41]. 

SHPRH overexpression increased the mutation frequency compared to both SHPRH F1632A overexpressing and control cells (Figure 3A), suggesting dysregulated DDT after SHPRH overexpression. 

In line with our data, it is reported that HLTF is more important for the regulation toward error-free DDT over UV-induced DNA damage than SHPRH, i.e., the mutation frequency was increased after HLTF knockdown but not after SHPRH knockdown [19]. The mutation spectra of *supF* reveal multiple differences between plasmids replicated in cells overexpressing wild type and APIM mutant versions of the RAD5 homologs. This further illustrates the impact of a direct interaction with PCNA for both HLTF and SHPRH (Figure 3B,C). 

### 2.5. Direct Binding of HLTF to PCNA Is Important for Error-Free DDT and/or DNA Repair

When analyzing the mutations in *supF* isolated from the different cells, we found mainly C to T transitions (corresponding G to A mutations on the transcribed strand) in plasmids from control cells (Figure 3D,G). DNA mutations resulting from UV exposure are usually ~80% C to T transitions [42], and, accordingly, C to T mutations are frequently found in skin cancers [43]. Thus, our results follow the expected mutation pattern from UV irradiation in vitro and in vivo. Overexpression of HLTF led to an increase of transitions from 83% (in control) to 90%, and this increase was mainly caused by an increase in tandem CC to TT mutations (Figure 3D). UV induced 6-4PPs are quickly repaired by NER, while CPDs are repaired at a slower rate. Therefore, the main amount of mutations received in our experiments likely arose from the bypass of CPDs (TT, TC, and CC). C to T transitions are the most frequent mutations found in both XPV cells (lacking POLη [44]) and normal cells. However, knockdown of POLη in mammalian cells does not change the amount of C to T mutations after UV irradiation [38,45]. This suggests that also other TLS polymerases than POLη can bypass CPDs leading to C to T transitions. Thus, whether the increase in CC to TT mutations is a result of HLTF mediated stimulation of POLη or stimulation of other TLS polymerases cannot be concluded based on our data.

Overexpression of HLTF F960A resulted in fewer transition mutations compared to HLTF overexpression, especially on the coding strand (26% versus 42% C to T mutations, and 7% versus 12% CC to TT mutations), and an overall increase in mutations on the transcribed/noncoding/template strand (Figure 3D). At the same time, HLTF F960A overexpression led to an increase in mutation frequency (Figure 3A). Since transcription-coupled NER is error-free and probably repairs part of the lesions on the transcribed strand, HLTF F960A overexpression is suggested to either dysregulate NER and/or other error-free repair/bypass processes, for example, fork reversal or TS. Thus, our results suggest that a direct interaction between PCNA and APIM in HLTF is important for error-free TLS and/or DNA repair regulated by HLTF. 

### 2.6. SHPRH Overexpression Stimulates Error-Prone TLS 

Overexpression of SHPRH led to an increase of mutation frequency by 38% (Figure 3A) and an increase in transitions, mainly due to a higher amount of C to T transitions (51% compared to 43% in control) (Figure 3D). Mutations at position 164 and 172 further illustrate this higher amount of mutations at Cs in SHPRH overexpressing cells (Figure 3E). POLκ has been shown to be activated by SHPRH overexpression [19], and thus the observed C to T transitions could be mediated by POLκ. In the presence of SHPRH F1632A overexpression, we observed a mutation spectrum that was more similar to the control than to SHPRH wild type overexpression. This might be caused by lower nuclear level of SHPRH F1632A, because function and nuclear localization of SHPRH was dependent on a direct APIM-mediated PCNA interaction (Figure 3B). 

Since HLTF and SHPRH are reported to be partly competitive [19,38], the changes in mutation patterns observed after overexpression of both wild type and APIM mutants of SHPRH and HLTF result from a disturbance of the HLTF/SHPRH ratio. 

### 2.7. Reduced Level of Putative Transcribed Strand Mutations after Overexpression of HLTF and SHRPH

Due to the nature of UV lesions (mainly occurring at C and T bases) and the fact that we sequenced the coding strand only, we categorized mutations into mutations which with high probability occurred on the coding strand versus transcribed strand. We found a lower fraction of mutations on the coding strand (44% vs. 56% on transcribed strand) in *supF* isolated from control cells (Figure 3D). This could be explained by transcription coupled NER repairing lesions on the transcribed strand and/or by a higher TLS rate on the coding strand. When HLTF or SHPRH were overexpressed, the mutations on the coding strand decreased to 40%. This could be explained by increased TLS on the coding strand (e.g., POLη or κ) or decreased error-free repair/bypass on the transcribed strand (e.g., TS, NER or error-free TLS). Overexpression of HLTF F960A resulted in an increased amount of G to A (48%) transitions compared to overexpression of wild type HLTF (29%) (Figure 3D). In addition, mutations on the transcribed strand increased from 40% for HLTF to 55% for HLTF F960A. The number of mutations at position 109 (mutations on coding strand CC) and 122 (mutations on transcribed strand CC) in the *supF* gene (Figure 3F) further illustrates the large difference between HLTF and HLTF F960A overexpression on coding and transcribed strand mutations. The same trends, although less pronounced, were detected for SHPRH wild type versus SHPRH F1632A, in spite of the low nuclear concentration of the mutated proteins. SHPRH overexpression resulted in a shift from G to A (30%) mutations to C to T (51%) mutations in comparison to SHPRH F1632A overexpression (37% and 46%, respectively) (Figure 3D), and differences in the mutation spectra especially in the amount of mutations at positions 124 and 164 and in the type of mutation, e.g., at position 122 (Figure 3B,E). In total, these results further support that APIM in HLTF as well as in SHPRH are functional PCNA interacting motifs, which seem to be especially important for reducing mutations on the transcribed strand. 

In conclusion, the results presented here show that both mammalian RAD5 homologs, HLTF and SHPRH, have functional APIM sequences and that their direct interactions with PCNA are important for the regulation of the DDT pathways. Our data also show that increased levels of wild type HLTF and SHPRH disturb the balance of error-free and error-prone DDT pathways, suggesting that their expression levels are normally strictly regulated.

## 3. Material and Methods

### 3.1. Expression Constructs 

pEGFP-C2-HLTF and pEGFP-C2-SHPRH, the pSP189 reporter plasmid and *E. coli* strain MBM7070 described in [19], were kind gifts from Professor Karlene Cimprich, Department of Chemical and Systems Biology, Stanford University, USA. HLTF was sub-cloned into pEYFP-C1 vector (YFP-HLTF) and site-specific mutations at position F960A in HLTF, and F1632A in SHPRH were generated as described in [24]. The wild type APIM sequences from HLTF (aa 959–963, KFIVK) and SHPRH (aa 1631–1635, RFLIK), as well as the corresponding APIM mutant (F2A), were cloned as fusions with CFP or YFP, respectively, by using pECFP-N1 and pEYFP-N1 vectors with mutated ATG, similarly to RWLVK-CFP [24]. CFP-PCNA and HcRed-PCNA have previously been described [46]. 

### 3.2. Cell Lines 

HEK293 and HEK293T (ATCC: CRL-1573, CRL 11268, respectively) were cultured in D-MEM (4.5 g/L glucose; Sigma-Aldrich, Saint Louis, MO, USA). Media were supplemented with 10% fetal bovine serum (FBS; Sigma-Aldrich), 2.5 µg/mL of Fungizone ^®^ Amphotericin B (Gibco, Thermo Fischer Scientific, Waltham, MA, USA), 1 mM of l-Glutamine (Sigma-Aldrich), and antibiotic mixture containing 100 µg/mL of penicillin and 100 µg/mL of streptomycin (Gibco). The cells were cultured at 37 °C in a 5% CO_2_ humidified atmosphere. 

### 3.3. SupF Assay 

The SupF mutagenicity assays were performed, essentially, as previously reported [19]. Briefly, the reporter plasmid pSP189 was irradiated with 600 mJ/cm^2^ UVB (312 nm), with UV lamp Vilber Lourmat, Bio Spectra V5. HEK293 cells were transfected with constructs of interest and UVB- irradiated pSP189 (including plasmids not exposed to UVR as controls), using X-treme GENE HP transfection reagent according to manufacturer protocol (Roche diagnostics); at least 3 biological replicas were conducted (3 transfections). Cells were harvested after 48 h for both isolation of plasmid and Western analysis. Isolated plasmids were DpnI (NEB, Ipswich, MA, USA) restriction digested to exclude original bacterial plasmids, in order to continue with only replicated plasmids. Isolated plasmids were transformed into *E. coli* MBM7070 cells and plated on indicator X-gal/IPTG/Amp agar plates. Blue/White screening was performed, and mutation frequency (white/blue colonies) was calculated for the different samples for several transfections (at least 3 replica). White and light-blue colonies were picked for re-streaking and DNA sequencing of s*upF* gene. Colonies that did not show a mutation in the sequencing results were afterward excluded, and the mutation frequency was recalculated.

### 3.4. Imaging 

Live cell imaging of HEK293 transfected with pGFP-SHPRH, pGFP-SHPRH F1632A, pYFP-HLTF, pYFP-HLTF F960A, pHcRed-PCNA, and YFP/CFP constructs of the APIM/mutated APIM in HLTF or SHPRH were performed 24 h after transfection, using a Zeiss LSM 510 Meta laser scanning microscope equipped with a Plan-Apochromate 63× /1.4 oil immersion objective. GFP and CFP were excited/detected at λ = 458 nm/BP470–500 nm. YFP was excited/detected at λ = 514 nm/BP530–600 nm, and HcRed was excited/detected at λ = 543 nm/LP615 nm, using consecutive scans. The thickness of the scanned optical slices was 1 µm. 

Live cell imaging of HEK293T transfected with pYFP-HLTF or pYFP-HLTF F960A together with pKFIVK-CFP or pRWLVK-CFP and pHcRed-PCNA was performed as described above, but YFP was detected at BP535–590 nm in these experiments. Fluorescence intensities were measured in cells with comparable expression levels of YFP and CFP, using the imaging processing software Fiji (ImageJ) version 1.06.2016. Average intensity within an area of interest (foci) was measured and divided by average intensity in the nucleus outside foci. 

For SupF assay transfection control, HEK293 cells were transfected with proportional amounts of transfection mix as used in the SupF assay. Live cell imaging was performed 48 h after transfection using Zeiss LSM 510 Meta laser scanning microscope as described above, in order to evaluate the transfection efficacy.

### 3.5. Measurement of Fluorescence Intensities After APIM-Peptide or Leptomycin B Treatment

HEK293 cells were transfected with pGFP-SHPRH, pGFP-SHPRH F1632A, pYFP-HLTF, or pYFP-HLTF F960A, using X-treme GENE HP transfection reagent according to manufacturer protocol (Roche diagnostics, Basel Switzerland). After 24 and 30 h, the cells were treated with 8 μM of APIM-peptide (Innovagen, Lund, Sweden), 8 μM of APIM-peptide (Innovagen), or 30 ng/mL of Leptomycin B. Not treated is the negative control. Forty-five minutes after the treatment, images were taken, and the fluorescence intensity (mean) in the region of interest, covering at least 200 pixels, using Zeiss LSM 510 software, were measured.

### 3.6. Preparation of Cross-Linked Cell Extracts

HEK293T cells were transfected with the construct of interest and treated with 50 μM of methyl methanesulfonate (MMS) the next day. Forty-eight hours after MMS treatment, intact cells were gently cross-linked with 0.2% formaldehyde, as previously described [24]. Cells were lysed by using 3× PCV M-PER Mammalian Protein Extraction Reagent (Thermo Scientific, Waltham, MA, USA), PIC2 (10 μL/mL buffer), and PIC3 (10 μL/mL buffer) (Sigma Aldrich), and complete protease inhibitor (20 μL/mL buffer) (Roche) for 1 h at 4 °C and sonicated in a sonication water bath 5 rounds of 30 s on 30 s off (Picoruptor SA, Diagenode). Then, 1 μL of Omnicleave was added to 100 μL PCV. The lysate was cleared by centrifugation for 10 min, at 16000× *g*, 4 °C. 

### 3.7. Immunoprecipitation 

An in-house affinity-purified rabbit polyclonal antibody raised against GFP protein, which also recognizes YFP and CFP proteins (called α-GFP), was covalently linked to protein A paramagnetic beads (Invitrogen), according to instructions provided by New England Biolabs, Inc. Then, 750 μg of cell extract was incubated with 30 μL antibody-coupled beads and 0.3 mL of IP buffer (20 mM Hepes, pH 7.9, 1.5 mM MgCl2, 100 mM KCl, 0.2 mM EDTA, 10% glycerol, 1 mM DTT, 10 mM Na-butyrate, 0.1 mM NaVO3, and 1x complete protease inhibitor) under constant rotation, at 4 °C, overnight. The beads were washed three times with 500 μL of 10 mM Tris-HCl pH 7.4, with a 5 min incubation on ice in between each wash. The beads were resuspended in 30 μL of NuPAGE (Invitrogen) loading buffer, including DTT (final concentration 0.1 M), and heated for 30 min at 95 °C, to reverse crosslinks before the samples were run on 4–12% Bis-Tris-HCl (NuPAGE) gels. Then, 100 μg cell extract was used for input. The primary antibodies α-GFP (ab290, Abcam) and α-PCNA (sc-56, Santa Cruz Biotechnology), as well as the secondary antibodies IRDye 800CW (Goat Anti-Rabbit) and IRDye 700RD (Goat Anti-Mouse) (LI-COR Bioscience), were diluted in 5% dry milk in PBS, and the proteins were visualized and quantified by using the Odyssey Imager.

## Figures and Tables

**Figure 1 ijms-21-00693-f001:**
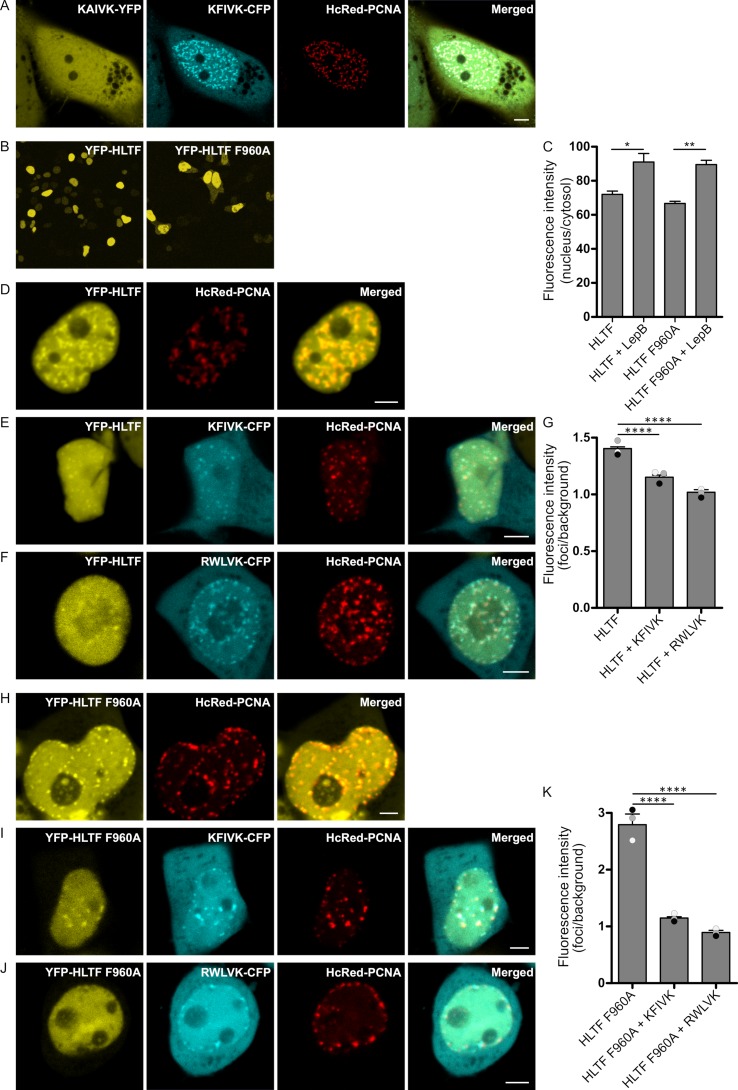
APIM in HLTF is a functional PCNA interacting motif. (**A**) Overexpressed KAIVK-YFP (mutAPIM in HLTF), KFIVK-CFP (APIM in HLTF), and HcRed-PCNA. (**B**) Overview of subcellular localization of YFP-HLTF and YFP-HLTF F960A, and (**C**) quantification of their nuclear localization in cells treated with Leptomycin B (LepB; 30 ng/mL, 45 min) (*n* = >90 cells per sample). (**D**) Overexpressed YFP-HLTF and HcRed-PCNA; (**E**) YFP-HLTF, KFIVK-CFP (APIM of HLTF) and HcRed-PCNA; and (**F**) YFP-HLTF, RWLVK-CFP, and HcRed-PCNA. (**G**) Quantification of YFP-HLTF foci intensities alone (*n*(number of foci) = 132) and with co-transfection of KFIVK-CFP (*n* = 76) or RWLVK-CFP (*n* = 59) from three biological replica depicted in white, gray, and black dots. Bars represent averages. (**H**) Overexpressed YFP-HLTF F960A and HcRed-PCNA; (**I**) YFP-HLTF F960A, KFIVK-CFP (APIM of HLTF), and HcRed-PCNA; and (**J**) YFP-HLTF F960A, RWLVK-CFP, and HcRed-PCNA. (**K**) Quantification of YFP-HLTF F960A foci intensities alone (*n* = 84) and with co-transfection of KFIVK-CFP (*n* = 103) or RWLVK-CFP (*n* = 98) from three biological replica depicted in white, gray, and black dots. Bars represent averages. Quantifications in G and K are based on at least 10 different images per confocal dish/sample in cells with similar protein expression. Foci intensity quantifications were done by using processing software Fiji (ImageJ). Two-sided Student’s *t*-test, * *p* < 0.05, ** *p* < 0.01, **** *p* < 0.0001. All images are from live cells. Scale bar = 5 μm.

**Figure 2 ijms-21-00693-f002:**
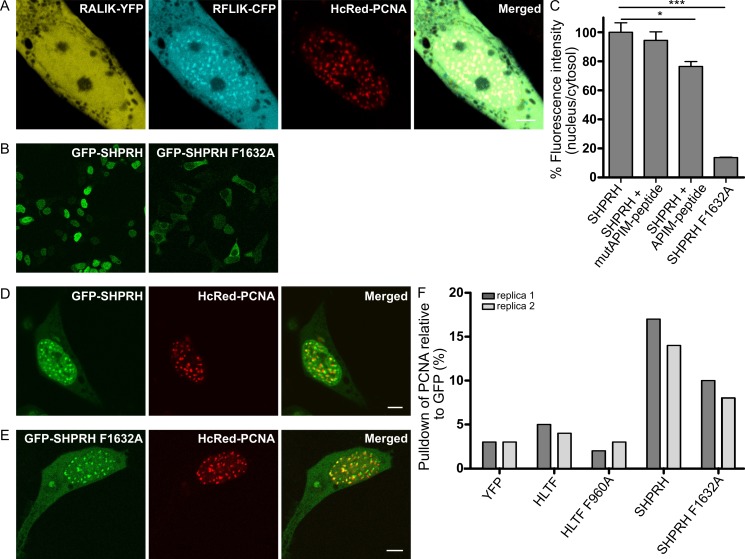
SHPRH localization in the nucleus is dependent on APIM. (**A**) Overexpressed RALIK-YFP (mutAPIM in SHPRH), RFLIK-CFP (APIM in SHPRH), and HcRed-PCNA. (**B**) Overview of subcellular localization of GFP-SHPRH and GFP-SHPRH F1632A, and (**C**) Quantification of nuclear localization of GFP-SHPRH (*n*(number of foci)= 484) and GFP-SHPRH F1632A (*n* = 123), and GFP-SHPRH after treatment with an APIM peptide (*n* = 247) or mutAPIM-peptide (*n* = 319), average of three biological replica, normalized to untreated control, two-sided Student’s *t*-test, * *p* < 0.05, *** *p* < 0.001. (**D**) Overexpressed GFP-SHPRH and HcRed-PCNA and (**E**) GFP-SHPRH F1632A and HcRed-PCNA. All images are from live cells. Scale bar = 5 μm. (**F**) Quantification of PCNA level pulled down by anti-GFP from YFP-HLTF, YFP-HLTF F960A, GFP-SHPRH, and GFP-SHPRH F1632A transfected cells after weak cross-linking and MMS treatment. Level of PCNA is given as % of total GFP-protein pulled down. Two independent biological replicas are shown.

**Figure 3 ijms-21-00693-f003:**
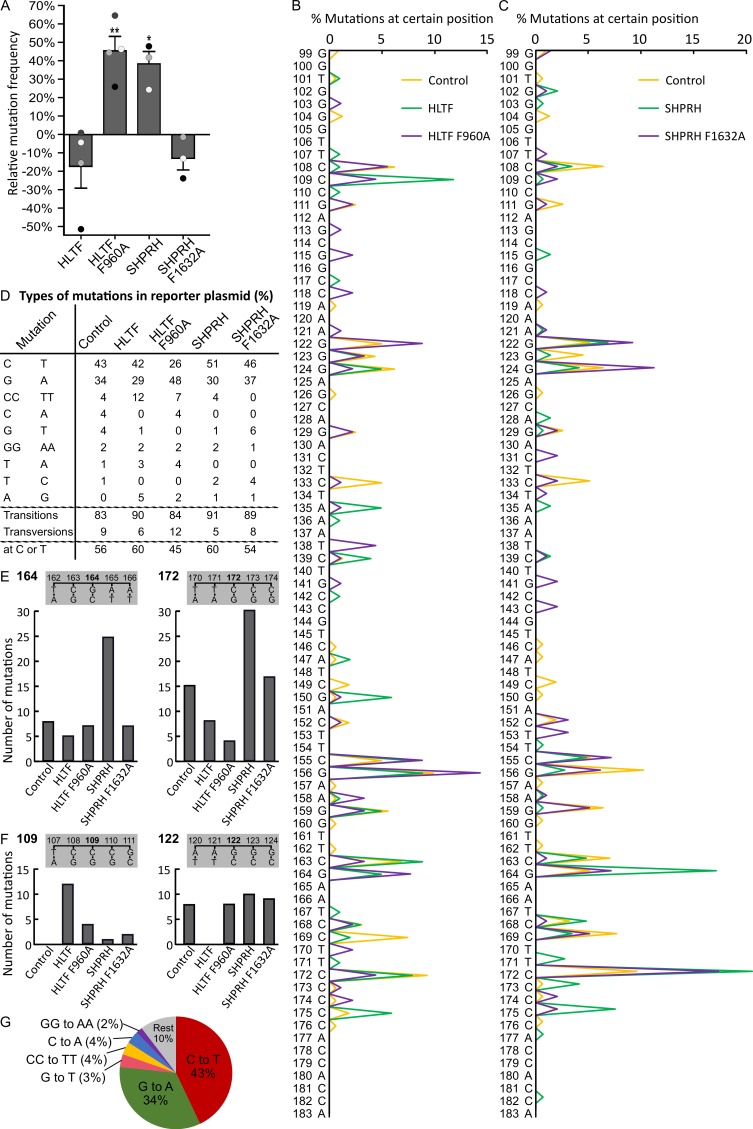
Mutation of APIM in HLTF and SHPRH results in differences in mutation frequencies and mutation spectra. (**A**) Mutation frequencies after overexpression of HLTF, HLTF F960A, SHPRH, or SHPRH F1632A, together with SupF reporter plasmid pSP189, relative to control (reporter plasmid only) in HEK293 cells from three or four biological replica (depicted as white, grey, and black dots). DNA from each biological replica was transformed until obtaining 1000–4000 colonies per replica. The total number of colonies counted are: HLTF (*n* = 14700), HLTF F960A (*n* = 13321), SHPRH (*n* = 11183), SHPRH F1632A (*n* = 14709), and control (*n* = 17560). Two-sided Student’s *t*-test relative to control, * *p* < 0.05, ** *p* < 0.01. (**B**,**C**) Mutation spectra received from sequencing mutant colonies from (A). HLTF (*n* = 102), HLTF F960A (*n* = 91), SHPRH (*n* = 152), SHPRH F1632A (*n* = 102), and control (*n* = 163). (**D**) Quantification of different mutations using sequencing results from (B,C). Mutations with prevalence ≥ 2% are shown. Mutations at T or C bases (putative coding strand mutations) in *supF* (**E**) Mutations at position 164 and 172 in *supF* isolated from cells overexpressing HLTF, HLTF F960A, SHPRH, or SHPRH F1632A compared to control. (**F**) Mutations at position 109 and 122 in *supF* isolated from cells overexpressing HLTF, HLTF F960A, SHPRH, or SHPRH F1632A compared to control. (**G**) Distribution of mutations using sequencing results from (B,C) received from reporter plasmids isolated from control cells.

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
