# Peer review of "Helicase-Like Transcription Factor HLTF and E3 Ubiquitin Ligase SHPRH Confer DNA Damage Tolerance through Direct Interactions with Proliferating Cell Nuclear Antigen (PCNA)"

_ijms, 2020, doi:10.3390/ijms21030693_

Round 1

Reviewer 1 Report

Seelinger and Otterlei questioned the functional significance of the AlkB homolog 2 PCNA interactive motifs (APIMs) identified within two RING domain E3 ubiquitin ligases HLTF and SHPRH for the DNA damage tolerance in human cells. The authors show that the respective APIM peptides (but not their F2A mutants) fused to fluorescent proteins co-localise with PCNA at the replication foci. Although the HLTF APIM mutant F960A retained co-localisation with PCNA, overexpression of the HLTF APIM peptide KFIVK (and even more its enhanced version RWLVK) efficiently outcompeted HLTF, thus suggesting that the interaction is partly mediated by the APIM sequence. The SHPRH APIM mutant has lost nuclear localisation, suggesting that the interaction with PCNA may be important for SHPRH retention in the nucleus. To confirm physical interaction with PCNA, authors further performed pulldown assays under the mild corisslinking conditions; however the results are, in my opinion, ambiguous (see comment 2 below). Importantly, overexpression of either HLTF F960A or SHPRH (wt) resulted in a significant mutation gain in the SupF assay, suggesting that balanced expression of both proteins affects the choice between the error-free and error-prone damage tolerance pathways. This is interesting and well performed piece of work and the evidence presented in the manuscript is worthy of publication, even though it is cannot be excluded based on the presented data that HLTF and SHPRH act in a DNA repair pathway as an alternative (or in addition) to the proposed damage tolerance mechanism.

Major points

1) A feasible and potentially very informative experiment would be to assess functional consequences for a distrupted HLTF-PCNA interacion using a DNA damage tolerance-relevant endpoint (e.g. the SupF mutation assay in the presence of overexpressed APIM peptides). Such experiment would provide additional support in favour of modulation of a damage tolerance mechanism (via a PCNA-interaction mechanism) rather than the nucleotide excision repair.

2) It is necessary to show the GFP and YFP controls in the Western blots shown in the Supplementary material (Figure S1). It would be preferable to show data relevant to the pulldown (currently Figure S1 and quantification in Figure 2F) in the same Figure. It is not clear whether YFP-HLTF yields a significantly higher PCNA pulldown than YFP alone. It is also not clear how SHPRH F1632A mutant localised outside of the nucleus could pull down significant amounts of PCNA. Besides, massive degradation of the HCTF and SHPRH fusion proteins (and expecially unequal degradation of HLTF and HLTF F960FA) makes a valid interpretation of the pulldown data difficult or impossible. Please reflect in the text that pulldown of PCNA by YFP-HLTF and GFP-SHPRH does not provide a conclusive indication whether APIMs are essential for the interactions. Is there a reason for which MMS was chosen as a damaging agent in the puldown assay. Would not UV be preferable, as mutation phenotypes were subsequently documented for UV-damaged DNA?

3) It is unclear why overexpression of HLTF F960A causes increased mutation frequency in the SupF assay. Is there any evidence that HLTF F960A disrupts interaction of native HLTF with PCNA? Since the F960A mutant co-localise with PCNA at the replication foci (Figure 1H), the simplest explanation would be that HLTF F960A itself promotes a mutagenic DNA damage tolerance mechanism. An experiment with overexpressed APIM peptide (as suggested in comment 1) may clarify this issue.

Other points and corrections

4) If results of SupF assay are interpreted correctly, association of HLTF and SHPRH with PCNA at replication forks as well as pulldown of PCNA should be modilated by UV. Is it possible to provide such evidence?

5) Was PCNA ubiquitylation analysed upon overexpression of HLTF and SHPRH?

6) Co-localisation of HLTF with HCRed-PCNA (Figure 1D and 1H): YFP-HLTF is hard to compare with YFP-HLTF F960A, because YFP-HLTF displays a much higher signal level in the nucleus. Please comment on the expression levels of two proteins or pick a representative image. Ideally, allignment of the YFP and HcRed peaks should be shown in a cross-nuclear sections. Similarly, GFP-SHRPH and GFP-SHRPH F1632A (Figure 2D and 2E) seem to localise in proximity to HCRed-PCNA, however do not precisely overlap.

7) It would be helpful to provide more specific information about HLTF and SHPRH in the title: “Roles of human Rad5 homologs HLTF and SHPRH…” or “Roles of the RING domain ubiquitin ligases HLTF and SHPRH…”

8) The statement “stimulation of error-free bypass in presence of HLTF overexpression” (abstract, lines 16-17) does not accurately reflect the result shown in Figure 3A, where the effect of wt-HLTF is mild (<20%) and apparently not significant.

9) HEK293, HEK293T and HeLa cell lines are listed in Materials and Methods (line 293). By what criteria were different cell lines chosen for specific types of experiments? What is 4.5 g/L (line 294)?

Page 1 (lines 13): “DTT” should read “DDT”

Page 1 (lines 16): delete the word “both”

Page 2 (line 56) “suggested as” should read “regarded as candidate” or similar

Page 6 (line 168): delete the word “therefore”

Page 10 (lines 288 and 296) “was” should read “were”

Caption to Figuire S1: “upper panel” should read “left panel”; “lower panel” should read “right panel”

Author Response

Point-to-point rebuttal to ijms; 680044, please see attachment

Reviewer 2 Report

The authors present a functional investigation of the RAD5-like mammalian ubiquitin ligases SHPRH (SNF2, histone-linker, PHD and RING finger domain-12 containing helicase) and HLTF (helicase-11 like transcription factor) that are involved in DNA damage tolerance mechanisms of replicating S-phase cells. Both are presumable tumor suppressor proteins that play roles in genomic caretaker mechanisms whose failure contributes to cancer development.

The authors used state of the art cell biological and mutational analyses and arrive at the conclusion that SHPRH and HLTF are both PCNA-interacting proteins whose stoichiometry in cells is required to faithfully direct DDT mechanisms that are dependent on PCNA on replicating DNA. They obtained evidence that the PCNA interaction is dependent on the motif APIM in both HLTF and SHPRH. Furthermore, they observe that binding of HLTF and SHPRH to PCNA contributes to DNA damage tolerance in cells with UV-damaged DNA.

The observations made are important and the manuscript is well written and displays convincing data. It is, however, packed with abbreviations and the list of these presented at the end of the manuscript requires some more effort. In conclusion, I can recommend its publication in IJMS provided a few amendments are made.

Points:

What about the statistical comparisons of the data mentioned in lines 207+ ? Is an increase from 83 to 90% statistically significant? Same applies to line 218/219, is the 7% vs 12% CC to TT… significant? The authors should present repeat experiments and statistical analyses. 

Minor points

Line (L) 34: replace “…to originate” with “… to induce…”

L46: ‘…K164 on PCNA’ – may be replaced with “K164 of PCNA”

L 73: “…directly interacts…” replace with …interact…

L 118: Mention with Plugin of ImagJ was used for analysis.

L 125: “first and forth bar” – not necessary here.

L 145: It would be good for the non-initiated reader to mention at this point which DNA lesions are induced by MMS treatment.

L 168: A reference to the SupF assay would be good here. Same line: replace “therefore” with “thereby”.

Please extend the abbreviations list, since it’s not complete. Missing are for instance: APIM, PCNA, DTT, TS, TLS, NER, APIM, SupF, …

Author Response

Point-to-point rebuttal to ijms; 680044

Referee 2

The authors present a functional investigation of the RAD5-like mammalian ubiquitin ligases SHPRH (SNF2, histone-linker, PHD and RING finger domain-12 containing helicase) and HLTF (helicase-11 like transcription factor) that are involved in DNA damage tolerance mechanisms of replicating S-phase cells. Both are presumable tumor suppressor proteins that play roles in genomic caretaker mechanisms whose failure contributes to cancer development.

The authors used state of the art cell biological and mutational analyses and arrive at the conclusion that SHPRH and HLTF are both PCNA-interacting proteins whose stoichiometry in cells is required to faithfully direct DDT mechanisms that are dependent on PCNA on replicating DNA. They obtained evidence that the PCNA interaction is dependent on the motif APIM in both HLTF and SHPRH. Furthermore, they observe that binding of HLTF and SHPRH to PCNA contributes to DNA damage tolerance in cells with UV-damaged DNA.

The observations made are important and the manuscript is well written and displays convincing data. It is, however, packed with abbreviations and the list of these presented at the end of the manuscript requires some more effort. In conclusion, I can recommend its publication in IJMS provided a few amendments are made.

Points:

What about the statistical comparisons of the data mentioned in lines 207+ ? Is an increase from 83 to 90% statistically significant? Same applies to line 218/219, is the 7% vs 12% CC to TT… significant? The authors should present repeat experiments and statistical analyses. 

Our answer:

These results are based on all the experiments, i.e. from 3 biological replica with at least 4 transformations. Number of sequenced colonies are: HLTF (n=102), HLTF F960A (n=91), SHPRH (n=152), SHPRH F1632 (n=102), control (n=163). These numbers are given in the figure legend.

We have not examined/evaluated the differences between individual transformations or biological replica, because we sequenced unequal amounts of colonies from each  transformation (3 biological replica with more than 4 transformation per biological replica). Our goal was to sequence approximately an equal high amount of sequences for each group in order to get an overview of the mutations after UV irradiation and overexpression of proteins of interest. We think that our data containing a large number of sequences show clear trends. However, as we did it this was, we cannot say if it is significantly different between the replica or transformations. However, together with the other data, the mutation spectra clearly indicate that the APIM motifs in HLTF and SHPRH afffects the mutation spectra, and represents functional PCNA interacting motifs in these proteins.

Minor points

Line (L) 34: replace “…to originate” with “… to induce…”

Our answer:

We agree, and have made corrections in the text

L46: ‘…K164 on PCNA’ – may be replaced with “K164 of PCNA”

Our answer:

We agree, and have made corrections in the text

L 73: “…directly interacts…” replace with …interact…

Our answer:

We agree, and have made corrections in the text

L 118: Mention with Plugin of ImagJ was used for analysis.

Our answer:

We used the the imaging processing software Fiji (ImageJ) version 1.06.2016 for these analyzis.

L 125: “first and forth bar” – not necessary here.

Our answer:

We agree, and have made corrections in the text

L 145: It would be good for the non-initiated reader to mention at this point which DNA lesions are induced by MMS treatment.

Our answer:

We agree and have added “the DNA alkylating agent MMS” to this sentence.…

L 168: A reference to the SupF assay would be good here. Same line: replace “therefore” with “thereby”.

Our answer:

We have deleted therefore, but references to the SupF assay is given in material and methods, and therefore not included here.

Please extend the abbreviations list, since it’s not complete. Missing are for instance: APIM, PCNA, DTT, TS, TLS, NER, APIM, SupF,

Our answer:

We have added PCNA to the list, the rest was already covered.

Round 2

Reviewer 1 Report

The author have amended the critical points. The manuscript is now suitable for publication.

Author Response

I could not find any additional points to comment on from this reviewer